# Therapeutic Effects of the Pilates Method in Patients with Multiple Sclerosis: A Systematic Review

**DOI:** 10.3390/jcm11030683

**Published:** 2022-01-28

**Authors:** Gustavo Rodríguez-Fuentes, Lucía Silveira-Pereira, Pedro Ferradáns-Rodríguez, Pablo Campo-Prieto

**Affiliations:** 1HealthyFit Research Group, Department of Functional Biology and Health Sciences, Faculty of Physiotherapy, Galicia Sur Health Research Institute (IISGS), University of Vigo, 36005 Pontevedra, Spain; pcampo@uvigo.es; 2Physiotherapist, Private Clinic, 36210 Vigo, Spain; luciasilveirapereira@hotmail.com (L.S.-P.); pedroferradans@gmail.com (P.F.-R.)

**Keywords:** multiple sclerosis, pilates-based exercise, exercise therapy, neurorehabilitation, physical therapy modalities

## Abstract

The Pilates Method is a rehabilitation tool with verified benefits in pain management, physical function, and quality of life in many different physiotherapy areas. It could be beneficial for patients with multiple sclerosis (pwMS). The aim of the study was to summarize current evidence for the effectiveness of Pilates in pwMS. A comprehensive search of Cinahl, Scopus, Web of Science, PEDro, and PubMed (including PubMed Central and Medline) was conducted to examine randomized controlled trials (RCT) that included Pilates intervention in multiple sclerosis. The PEDro scale and the Cochrane risk-of-bias tool, RoB-2, were used to evaluate risk of bias for RCT. Twenty RCT (999 patients) were included. Ten were of good quality (PEDro), and seven had low risk of bias (RoB-2). Pilates improves balance, gait, physical-functional conditions (muscular strength, core stability, aerobic capacity, and body composition), and cognitive functions. Fatigue, quality of life, and psychological function did not show clear improvement. There was good adherence to Pilates intervention (average adherence ≥ 80%). Cumulative data suggest that Pilates can be a rehabilitation tool for pwMS. High adherence and few adverse effects were reported. Future research is needed to develop clinical protocols that could maximize therapeutic effects of Pilates for pwMS.

## 1. Introduction

Multiple sclerosis (MS) is a chronic autoimmune and inflammatory neurological disease that affects the myelinated axons in the central nervous system, characterized by neurological deterioration over time [1]. MS is the most common non-traumatic disabling disease in young adults [2]. It usually starts in early adult life, typically in the third decade [3], with most patients presenting with periodic neurological relapses [4], but the disease course is unpredictable [5].

MS is one of the most common diseases of the central nervous system (2.2 million people worldwide in 2016 data) [4]. MS cases are twice as high in women as in men [1], and it is more prevalent in North America, Western Europe, and Australasia [4].

MS shows several patterns: 80% of all cases are “relapsing-remitting” MS (RRMS), characterized by exacerbations and remissions, which can turn into “secondary-progressive” MS (SPMS), with progressive disability between attacks; 15% are cases of “primary-progressive” MS (PPMS), where there is a progressive disability from the beginning; and 5% are “progressive-relapsing” MS (PRMS), where the disease worsens gradually, but also presents outbreaks [5]. However, Lublin et al. [6] recommended reviewing these descriptions of the clinical course or phenotype of MS in 2014, suggested defining phenotypes based on disease activity (based on clinical relapse rate and imaging findings) and disease progression, and recommended removing the PRMS phenotype.

MS is characterized by a wide spectrum of symptoms, including cognitive dysfunction, optic neuritis, diplopia, sensory loss, muscle weakness, gait ataxia, loss of bladder control, spasticity, and excessive fatigue [1,4,7]. In addition, patients with multiple sclerosis (pwMS) present high risk of falling (fall rate of 56%) [8].

Physical exercise has been postulated as one of the non-pharmacological strategies of interest, due to its low cost and positive effects on the physical and mental health of the MS population [9,10,11]. Although historically, exercise was not recommended for pwMS due to fear of aggravating the disease [12], current evidence indicates that physical exercise is positive for managing symptoms, restoring function, optimizing quality of life, and facilitating activities of daily living [13,14,15,16,17,18]. Tallner et al. [19] and Pilutti et al. [20] suggest that physical activity has no significant influence on clinical disease activity.

Pilates is a method of physical exercise that focuses on core stability, strength, flexibility, posture, muscle control, breathing, and mind–body connection [21]. Nowadays, the method is an accepted rehabilitation tool, with verified benefits in pain management, physical function, and quality of life when used as an intervention in many different physiotherapy areas [22,23]. This therapeutic modality of the Pilates Method could provide improvement of functional impairment to pwMS.

This systematic review aims to provide an overview of the literature, and to analyse the therapeutic effects of Pilates in pwMS.

## 2. Materials and Methods

### 2.1. Search Process

This study was carried out according to the PRISMA (Preferred Reporting Items for Systematic Reviews and Meta-Analyses) guidelines [24]. The search strategy was designed to find research studies providing information on the therapeutic effects of Pilates in pwMS. Seven electronic databases were used for the search (Cinahl, Scopus, Web of Science, PEDro, and PubMed (including PubMed Central and Medline)), up to July 2021, using the key words and Boolean operators “Multiple Sclerosis” AND “Pilates” OR “Pilates-based exercises” OR “Pilates exercise” OR “Pilates training”.

### 2.2. Selection Procedure and Eligibility Criteria

Two reviewers (G.R.-F. and L.S.-P.) independently selected trials for inclusion using predetermined inclusion criteria. First, we screened by titles and abstracts. Second, we acquired the full text of the remaining citations, and read each one to determine eligibility. In all cases, we resolved any disagreements about trial inclusions by consensus among reviewers, and consulted a third reviewer (P.F.-R.) if disagreements persisted.

The selection criteria included randomised controlled clinical trials (RCTs) written in English, Portuguese, or Spanish up to July 2021.

### 2.3. Data Extraction

The information in each study regarding purpose, characteristics of the sample, type of intervention and characteristics of Pilates intervention, dropouts, adherence/attendance, study variables, assessments, findings, and adverse effects of the Pilates intervention was recorded in a data log grid by one author (G.R.-F.). The information was subsequently independently revised by another three authors.

### 2.4. Assessment of Methodological Quality

To assess the methodological quality of the RCTs, the PEDro scale [25] was applied by two authors independently (G.R.-F. and P.C.-P.). Whenever discrepancies emerged, a third author was requested (P.F.-R.). The suggested cut-off points for categorizing studies by quality with the PEDro scale were as follows: excellent (9–10), good (6–8), fair (4–5), and poor (<3) [26].

In addition, an assessment of the risk of bias was carried out with the Cochrane risk-of-bias tool for randomized trials, RoB 2 [27]. This tool is structured into five bias domains: bias arising from the randomisation process, bias due to deviations from intended interventions, bias due to missing outcome data, bias in measurement of the outcome, and bias in selection of the reported result. In addition, there is an overall risk-of-bias judgment that generally corresponds to the worst risk of bias in any of the domains. Risk-of-bias judgment for each domain and overall can be “low”, “some concerns”, or “high”.

## 3. Results

There were 204 articles initially identified through database searches. After duplicate titles were removed, 138 studies remained. Another 93 potential articles were removed after the title and abstract review. Forty-five full texts were reviewed, and twenty articles [28,29,30,31,32,33,34,35,36,37,38,39,40,41,42,43,44,45,46,47] were used in the analysis based on inclusion and exclusion criteria. Figure 1 shows the PRISMA selection process flow chart [24].

Table 1 shows a summary of the main findings of the reviewed studies, and more extensive information is presented in Appendix A. Table 2 summarizes the main features of the Pilates intervention, and Table 3 describes the methodological characteristics of the studies (PEDro scale and sample size calculation). Finally, Figure 2 shows risk of bias of the reviewed studies.

## 4. Discussion

The purpose of this review was to identify the possible therapeutic effects of Pilates for pwMS. These were the main variables studied, as well as the influence of Pilates on each of them.

### 4.1. Balance

Balance was studied in 14 papers [28,30,34,36,37,38,40,41,42,43,44,45,46,47]. In each of these, one of the following scales or assessment methods was used: Timed Up and Go Test [28,30,34,38,40,41,42,44,45,46], Berg Balance Scale [29,36,40,42,44,45,46], Activities-Specific Balance Confidence Scale [34,41,43,46], Functional Reach Test [30,40], Balance Platform [28,37], Falls Efficacy Scale International [34], Fullerton Advanced Balance Scale [38], Four Square Step Test [40], Single-leg Stance [41], Trunk Impairment Scale [44], and Six-and-Spot Step Test [47]. In all 14, Pilates yielded significant improvements post-intervention. The studies of Hosseini Sisi et al. [45] and Marandi et al. [47] only compared the results for this parameter with other therapies (rebound therapy [45] and aquatic therapy [47]), and did not report significant differences between interventions. In contrast, the study by Gheitasi et al. [30], with solid methodological quality (7 on PEDro scale), found significant improvement from Pilates compared with usual physician care; and Duff et al. [38] (7 on PEDro scale), which compared Pilates with 1-h of massage per week, found significant improvement from Pilates in balance and gait. 

These findings are in line with other reviews [48]. As such, balance seems to be an important issue in pwMS that can benefit from Pilates interventions.

### 4.2. Gait/Walking

This variable was assessed in 11 studies [28,31,32,34,36,37,38,40,42,43,44] using: 6 Minute Walk Test [32,34,36,37,38,40], 12-Item Multiple Sclerosis Walking Scale [31,34,40,43], 10 Meter Walk Test [31,36,43], Timed 25-Foot Walk [34,42,44], 2 Minute Walk Test (2MWT) [28,31,40], the walking section of the Patients’ Global Impression of Change Scale [31], and Rivermead Visual Gait Assessment [31]. Ten studies found significant improvements in gait post-intervention [28,31,32,34,36,37,38,40,43,44]. Of note are the studies by Arntzen et al. [31] (8 on PEDro scale), where there were significant differences in favour of Pilates with respect to standard physiotherapy care; and by Ozkul et al. [32] (8 on PEDro scale), where Pilates presented significant improvements in gait quality when compared with relaxation exercises carried out at home. Also of interest is the significant 2MWT improvement found by Güngör et al. [28], both for patients in the Pilates group under the supervision of a physiotherapist and those doing home-based Pilates training.

Gait is a variable intimately related to balance. Two papers focused on studying both parameters combined: Kalron et al. [40] and Fox et al. [43]. The two studies compared the results obtained with the Pilates intervention with those from standard physical exercise. In Kalron et al. [40], both groups improved, although no significant differences were found between them. In contrast, in Fox et al. [43], the group doing standardised exercises obtained significant post-intervention improvements compared with those undergoing the Pilates intervention. Therefore, although Pilates has positive effects on gait and balance in pwMS, it seems that they are no better than other modes of physical exercise.

### 4.3. Physical-Functional Conditions

Within this variable, we include those studies which assess muscle strength: (leg extension 1 RM [38], sit-ups test [41], modified push-ups test [28,41], quadriceps and hamstrings isokinetic strength [28], and hand held dynamometer [46]), core stability (curl-up test [28,34], plank hold test [38], side bridge test [28,41], trunk flexion test [28,41], prone bridge test [41], and Biering-Sorensen test [28]), physical performance (9-Hole Peg test [42,44]; and time to roll from right to left, lie/sit, sit/stand, and repeated sit/stand [42,44]), aerobic capacity (consumption of VO_2_ on treadmill, and Physiological Cost Index [33]), physical activity (accelerometer monitoring activity [31,38], Godin Leisure-Time Exercise Questionnaire [29,35]), and body composition [36]. The reported results suggest that intervention with Pilates could be a valid tool for improving strength [33,46], core stability [34,41], physical performance [42,44], aerobic capacity [33], and body composition [36] in pwMS. However, the heterogeneity of the assessment tests employed hampers data aggregation and direct comparison of the results.

### 4.4. Fatigue

Pilates improved fatigue significantly in pwMS in nine studies [28,29,32,35,36,37,41,42,44] of the ten that evaluated it [28,29,32,35,36,37,40,41,42,44]. Nevertheless, none of these found significant differences when compared with other interventions. Once again, different scales were used to evaluate fatigue: Modified Fatigue Scale [29,35,36,40,44], Fatigue Impact Scale [32,42], and Fatigue Severity Scale [28,37,41]. Pilates provides positive results, but whether it is better than other treatments remain unclear. Specifically, in the study by Kara et al. [42], the group doing aerobic exercises did obtain significant post-intervention improvements in fatigue, but the Pilates group did not. Nonetheless, this result needs to be interpreted with caution, because there were not significant differences between the groups, the sample size was small, and there were a lot of losses in the Pilates group post-intervention. Güngör et al. [28] also obtained significant improvements in fatigue in both the supervised Pilates training group and the home-based Pilates training group, but without differences between the groups, although there was a loss of 20% from the latter group, which could have altered the findings.

In summary, fatigue remains a poorly studied variable [4,7] despite being a widespread alteration in pwMS.

### 4.5. Quality of Life

Four of the studies evaluated this parameter [32,38,41,44]. Two scales were used: the Multiple Sclerosis Quality of Life-54 instrument [32,38,41], and the Multiple Sclerosis International Quality of Life Questionnaire [44]. Significant improvements were obtained for this variable in three of the four studies [32,41,44] in both the physical and mental sections of the scales; in one of which [44], the results were significantly better in the Pilates group than in the control group.

### 4.6. Cognitive/Psychological Function

Cognitive functions were analysed in four studies [32,34,42,44], using the following scales: Brief Repeatable Battery of Neuropsychological Tests [32], Brief International Cognitive Assessment for MS [34], and Paced Auditory Serial Addition Test [42,44]. All of them obtained significant post-intervention improvements in this parameter. Of interest are the studies by Ozkul et al. [32] and Abasiyanik et al. [34], which compared the Pilates intervention with doing exercises at home (relaxation in the case of Ozkul et al. [32]), and obtained significantly better results in this variable with the Pilates intervention. Although somewhat surprising, the results in these studies open the door to incorporating measurements of cognitive parameters in future work using Pilates, as has been done in others where exercise was also the base of the intervention [49,50,51,52,53].

On the other hand, Fleming et al. [29,35] assessed depression and anxiety in pwMS, and Küçük et al. [44] assessed depression. The results do not offer any clear direction: although in Fleming et al. [29], the home-based Pilates group obtained significant improvement in comparison with the control group with regard to depression and anxiety, in earlier work [35], the same author stated that the supervised Pilates group presented a significant worsening of anxiety symptoms with respect to the home-based Pilates group. In addition, Küçuk et al. [44] did not find significant improvements in depression following the Pilates intervention. It is possible that these results are due to other factors, such as the comorbidities or severity of MS, as Kara et al. [42] report that both the Pilates and aerobic exercise groups, despite the improvement in depression experienced, did not present values significantly better than healthy adults.

### 4.7. Attendance/Adherence

In general, for the studies presenting data on compliance by patients with the Pilates sessions, there is an average adherence to the treatment above 80–85% [31,32,34,35,37,38,46]. The outstanding levels of adherence, in addition to the clinical results, are one of the highlights of using Pilates interventions in pwMS. Nevertheless, in the study by Fox et al. [43], the Pilates exercise group only had 66% adherence compared with 84% and 92% in the groups doing standardized exercises or relaxation, respectively. Most of the papers do not state values for compliance [28,29,30,33,36,39,40,41,42,44,45,47], which is an important limitation. Adherence is usually linked to the patient’s motivation for the treatment offered [54]; hence, it is a relevant issue in a disease such as MS, a long-duration chronic illness requiring physical-functional conditions to be as stable as possible over time, to maintain the independence and autonomy of patients. Lack of adherence may reflect a deterioration in the fitness of pwMS, and greater expense in terms of healthcare and personnel resources.

### 4.8. Sample Characteristics

The population analysed in these studies comprised 999 pwMS, and 868 finished them (131 dropouts, 13.11%). In terms of age, patients in their third and fourth decades predominated [28,29,30,32,34,36,37,38,39,40,41,42,45,46] (only three of the studies [31,43,44] include patients aged over 60), and this is in line with the epidemiological data [3]. With regard to sex, the majority are women (602), compared with 226 men, which also matches the epidemiological data for the disease (3:1 women to men ratio [2,3]). Dropout rates may vary between sexes, as in the study by Bulguroglu et al. [41]. Surprisingly, in some of the studies, the sample consists exclusively of women [33,35,36,39,46,47] or men [30,45], which complicates cross-sex validation of the results.

The majority of the samples include MS of the RRMS type (86.76%), this being the most common form of MS [5]. In eight studies [29,30,35,41,44,45,46,47], the clinical state of the disease in the participants is unfortunately not specified. 

The degree of disability in the samples was quantified in most cases using the Expanded Disability Status Scale [28,30,31,32,33,34,36,37,39,40,41,42,43,44,45,46,47], with the average scores on this questionnaire being highly variable, tending normally to an average score of 4.5 [28,31,32,34,37,40,41,42,44,45,46,47]. Only the study by Banitalebi et al. [33] included patients with a score of up to 8 on this scale: patients need to use aids for walking once their score reaches 6. Three studies employed the Patient-Determined Disease Steps Scale [29,35,38] to measure the degree of disability. The lack of standard evaluations, combined with the fact that some studies do not specify the clinical type of MS [29,30,35,41,44,45,46,47], leads to important knowledge gaps that ought to be addressed in future research. The performance of participants in Pilates programs will determine the design of the exercises, and the therapeutic objectives intended for each type of MS. Amatya et al. [5] agree that it is key to analyse these aspects to offer more effective and specific multidisciplinary treatment for each pwMS.

### 4.9. Characteristics of the Pilates Interventions

The type of Pilates intervention is specified in the majority of the cases, with mat work being the preferred modality [28,29,30,31,32,34,35,36,37,38,39,41,42,43,44]. In Duff et al. [38], the Pilates group did sessions of mat work and fitness equipment, whereas in Bulguroglu et al. [41], a mat work group was compared with one using Pilates exercise machines, and with a control group doing relaxation and respiration exercises at home. The Pilates intervention on the mat is likely preferred for economy and space reasons, as well as for its convenience for group therapy sessions. However, in Bulguroglu et al. [41], although both modalities achieved significant post-intervention improvements, there is significantly greater improvement in the exercise machine group when looking at vertebral mobility using the Trunk Flexion Test. It would be useful for future studies to analyse whether working with machines offers greater benefits than mat work, in both therapeutic and cost-effectiveness terms. 

It is also challenging to evaluate the benefits offered by at-home video-guided Pilates interventions for pwMS. The studies [29,35] that presented this intervention offer encouraging results. The Pilates intervention guided by DVD obtained good results in relation to symptoms of anxiety, depression, and fatigue. It is unclear whether the DVD modality is a better choice than the supervised intervention, from both the therapeutic and cost-effectiveness points of view. It would be helpful to analyse DVD-guided Pilates intervention as a sole treatment, or as a complement to the work of health professionals, as well as the requirements necessary (for instance, workload recommended), or the potential options for tracking the workload or motivation of pwMS to continue with the Pilates program.

In the majority of the studies, the session duration ranges from 45 to 60 min [29,30,31,32,34,35,36,37,38,42,44,45,46,47], with weekly frequency mainly established at two [28,29,35,38,41,42,44,46] or three [30,31,32,33,36,37,39,45,47] sessions per week, except for three with one session/week [34,40,43]. Most interventions lasted 8 [28,29,34,35,36,37,39,41,42,44,45,46] or 12 weeks [30,33,38,40,43,47]. Because in most of the studies there was no long-term monitoring after the intervention, it is unclear whether the outcomes attained are maintained. Only the study by Arntzen et al. [31], the one with the shortest intervention (6 weeks), tracked outcomes up to 30 weeks. These results point to sustained benefits for gait in the Pilates group after 18 (walking speed, perceived limitations, and distance walked) and 30 weeks (distance walked). We propose to schedule follow-up assessments in order to define whether the effects of the Pilates persist, in addition to establishing, in cases where the treatment is suspended (such as for vacation), after how long the treatment ought to be resumed to avert a significant loss of the benefits achieved.

The sessions took place for individuals in six studies [28,29,35,40,41,43], and for groups in five studies [31,34,38,44,46], whereas in nine studies, was not specified [30,32,33,36,37,39,42,45,47]. Eight studies [30,33,38,40,41,45,46,47] do not provide details regarding the Pilates program applied. Future research should be more precise in how the interventions are described, as this would facilitate replication, comparison, and evolution of the protocols.

With regard to the session supervision, the professional in charge was a physiotherapist in 11 studies [29,31,32,34,37,40,41,42,43,44,46], 5 of which specified that a Pilates certification was held [34,40,41,43,46]. In six studies [30,33,36,39,45,47], the professional responsible for directing the sessions is not specified. From our point of view, the physiotherapist is the ideal person for conducting these interventions in patients, optimally with a Pilates specialisation, to guarantee more effective and safer sessions while following the guidelines of the method properly.

### 4.10. Adverse Effects and Dropouts

Whether the intervention had adverse effects is relevant for pwMS. Adverse effects during the intervention were specified in four studies [28,31,37,42], with a total of nine cases (five for exacerbation of symptoms, two for relapse, and two due to the work intensity). In seven studies [33,36,39,41,44,45,47], it is not specified whether there were any adverse effects, although there were some dropouts. Adverse effects should be reported, and the cause of dropping out should be clarified, as well as the possible link with undesired effects of the treatment, to provide assurance in future research with Pilates in pwMS. It is also relevant for the validity of the outcomes to be verified. As shown in Table 3, in nine studies [28,34,35,36,41,42,45,46,47], one key result could not be obtained in at least 85% of the initial sample, hindering attribution of the results to the intervention.

### 4.11. Methodological Quality of the Studies

Table 3 shows that the average score obtained by the studies was 5.5/10 on the PEDro scale. Following Foley et al. [26], a score of 9–10 means excellent quality, 6–8 is good, 4–5 is acceptable, and <4 points is poor. In our review, 10 studies [28,29,30,31,32,33,37,38,40,43] have a value ≥6 points (3 reach 8 points [31,32,43], and only 3 [42,45,47] have a score <4 points on this scale). Given the PEDro scale design, the principal limitation is that none of the studies were double-blind. It also seems important for future studies to include appropriate randomization of the sample (as reflected in domain 1 of Figure 2), and an analysis of the results by treatment intention, which would help to control the detection biases (domain 3, Figure 2). All these issues would enhance the internal validity of the studies and their bias control.

Table 3 shows that only nine studies [28,29,30,31,32,34,37,38,43] have incorporated a sample size calculation, thus facilitating extrapolation of their results. Future investigations should take this into account in their design, as this would bring external validity to the results, and enhance their ecological value. According to the present review, this issue can eminently be improved.

### 4.12. Limitations

The main limitation is the methodological quality of the studies: although acceptable in most cases, greater control over biases and larger population samples are required.

Furthermore, the lack of data about adverse effects, whether these are related to some of the dropouts, and the type of MS should be clarified, and could limit the scope of our conclusions if the disease status could be related to a higher number of dropouts. The tendency to analyse the influence of Pilates in pwMS with mild to moderate degree of disability is also relevant, so the studies included show limited information on the effects of Pilates in pwMS who are already starting to have significant problems with ADL and walking. Further, studies have often included patients with different types of MS or with different degrees of disability, and no systematic information is collected on drug treatment.

Another limitation is the lack of an adequate blind allocation, with masked assessors and intention-to-treat analyses. 

Overall, the need for standardisation is obvious, as it would allow future researchers to compare different studies, their results, and the effectiveness of the Pilates to choose the best approach for clinical use. Furthermore, the heterogeneity of the evaluation scales does not allow a reliable comparison between studies, or aggregation of the results to support the findings.

We suggest follow-up assessments in future studies to explore how long improvements last, and to propose suitable guidelines for the dichotomy between treatment period and pause period between treatment programs.

Finally, the exclusion of grey literature due to our study selection criteria could have led us to omit certain papers referring to our area of study, and whose results might have been of interest for this review.

## 5. Conclusions

The outcomes support the therapeutic use of Pilates in MS management. Our findings suggest that Pilates is a safe active treatment method for pwMS (few adverse effects), with high adherence (low dropout rate), and which can improve important parameters in the target population, such as balance, gait, physical-functional capacities, and even cognitive functions. Findings are fairly limited for other variables, as in the case of fatigue, quality of life, and psychiatric conditions such as depression or anxiety.

Additional high-quality RCTs with large enough population samples are needed to substantiate the advantages of Pilates as a tool for rehabilitation in pwMS.

## Figures and Tables

**Figure 1 jcm-11-00683-f001:**
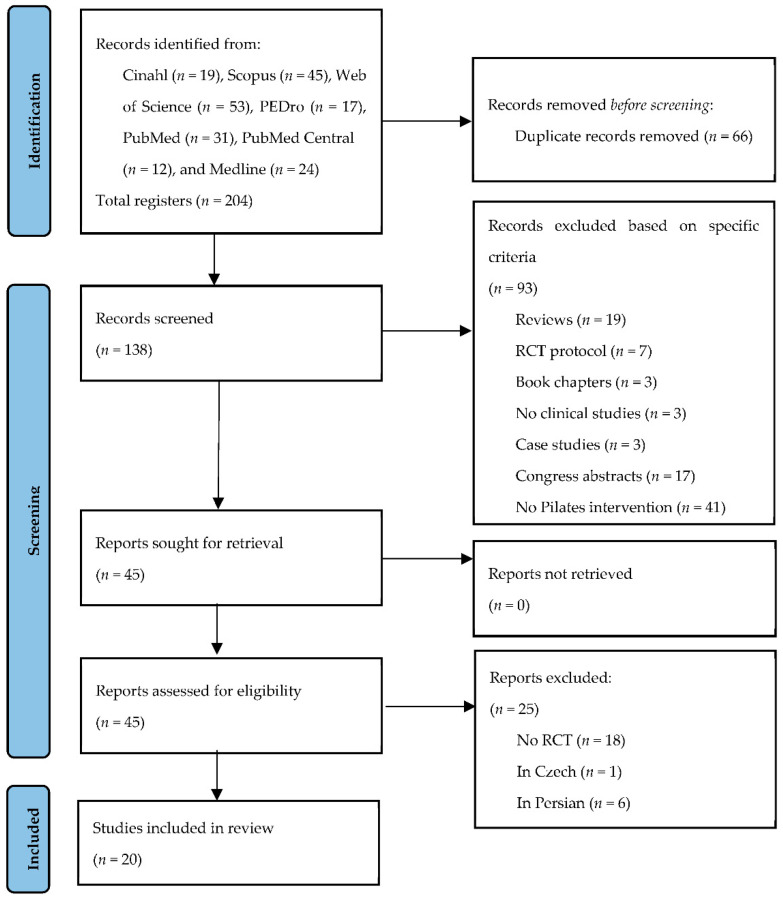
PRISMA flow chart of the study selection.

**Figure 2 jcm-11-00683-f002:**
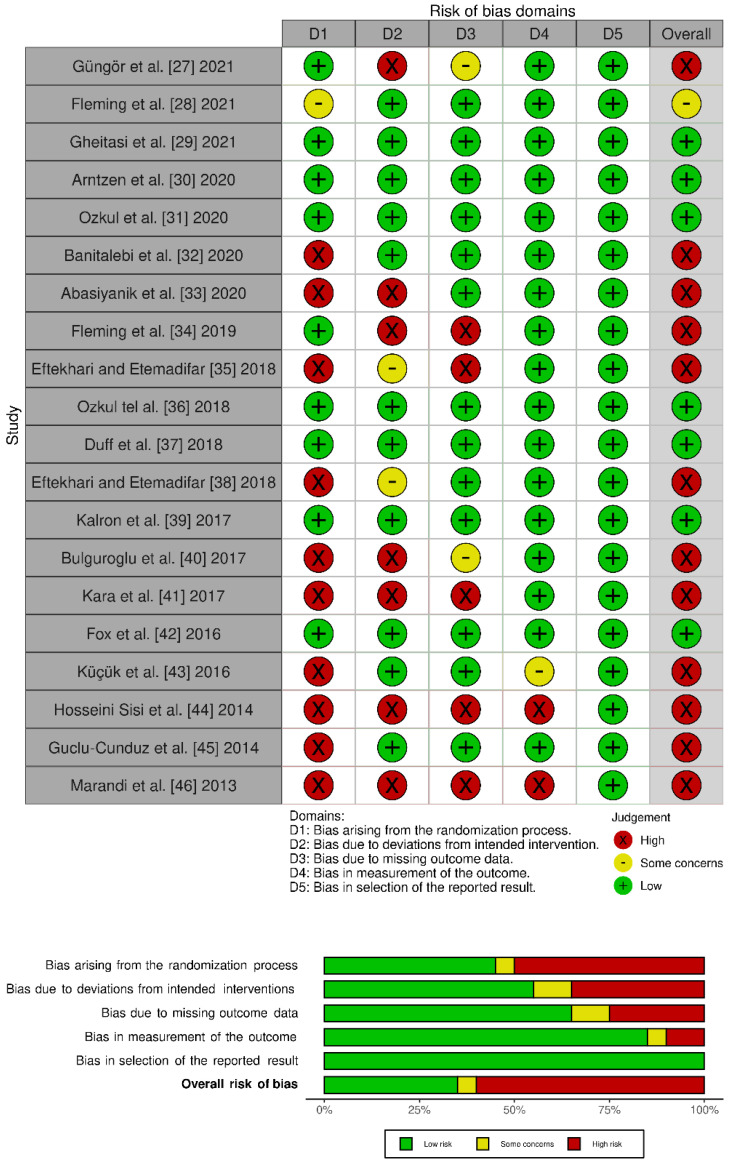
Risk of bias of the reviewed studies: (**up**) for each domain, (**down**) overall judgement.

**Table 1 jcm-11-00683-t001:** Main findings of the reviewed studies.

Variables	Main Findings with Pilates Method	Other Findings
Balance	Significant improvement [28,30,34,36,37,38,40,41,42,43,44,45,46,47] Significant improvement compared with physician care [30] or with 1-h massage [38]	No significant differences between Pilates and rebound therapy [45] or aquatic therapy [47]
Gait	Significant improvement [28,31,32,34,36,37,38,40,43,44] Significant improvement compared with standard physiotherapy care [31] or with home relaxation exercises [32]	No significant improvement [42] Significant improvement in standardized exercises group compared to Pilates group [43]
Physical-functional conditions	Significant improvement in muscle strength [28,33,41,42,46] Significant improvement in core stability [28,34,41] Significant improvement in physical performance [42,44] Significant improvement in aerobic capacity [33] Significant improvement in body composition [36]	
Fatigue	Significant improvement [28,29,32,35,36,37,41,42,44]	No significant improvement in post-intervention Pilates group, but there was in aerobic exercise group [42] (no difference between groups)
Quality of life	Significant improvement [32,41,44] Significant improvement compared with traditional exercises group [44]	No difference between Pilates + a 1-h massage therapy group and a 1-h massage therapy group [38]
Cognitive function	Significant improvement [32,34,42,44] Significant improvement compared with home relaxation exercises group [32] and with home exercises group [34]	
Psychological function	Significant improvement in depression symptoms [29] Significant improvement in anxiety [29] Significant improvement in depression symptoms and anxiety compared with wait-list group [29]	Significant improvement in home-based Pilates group compared with supervised Pilates group in anxiety symptoms [35] No significant improvement in depression symptoms [44]
Adherence	Average adherence to the treatment above 80–85% [31,32,34,35,37,38,46]	Lack of compliance values [28,29,30,33,36,39,40,41,42,44,45,47] Higher compliance in standardized exercises group or relaxation group than Pilates group [43]
Adverse effects and dropouts	9 cases of adverse effects [28,31,37,42] 62 dropouts in Pilates groups [28,29,32,33,34,35,36,38,39,41,46]	3 cases of adverse effects in no-Pilates groups [34,42] 45 dropouts in no-Pilates groups [29,31,34,35,36,37,39,40,42,43,47] 24 dropouts in unspecified group [41,44] Don’t know the dropouts [45] Don’t know the adverse effects cases [33,36,39,41,44,45,47]

**Table 2 jcm-11-00683-t002:** Pilates intervention characteristics of reviewed studies.

Study and Year	MS Type (Patients in Pilates Group)	Mean EDSS Score ± sd (Range)	Weeks	Session per Week	Session Duration (min)	Type of Pilates	Pilates Session (Number When Conducted in Group)	Professional	Adverse Events (Case)	Pilates Training Program
Güngör et al. [28], 2021	RRMS (34)/SPMS (8)	(1–5.5)	8	2	60–75	Floor mat work	Individual	Physiotherapist	No	Yes (in Appendix A)
Fleming et al. [29], 2021	NA	<3 (PDDS)	8	2	60	Floor mat work	Individual	Certified Pilates instructor	No	Yes (in a previous study)
Gheitasi et al. [30], 2021	NA	4.6 ± 1.6 (3–5)	12	3	60	Floor mat work	Unclear	NA	No	No
Arntzen et al. [31], 2020	RRMS (32)/PPMS (5)/SPMS (2)	2.45 ± 1.65 (1–6.5)	6	3	60	Floor mat work	Group (3)	Neurological physiotherapists	Yes (1)	Yes (in a previous study)
Ozkul et al. [32], 2020	RRMS (17)	1.50 ± 0.77 (<4)	8	3	60	Floor mat work	NA	Physiotherapist	No	Yes
Banitalebi et al. [33], 2020	RRMS (47)	23 (0–4) + 13 (4.5–6) + 11 (6.5–8)	12	3	15/100	NA	NA	NA	NA	No
Abasiyanik et al. [34], 2020	RRMS (14)/SPMS (2)	3.06 ± 1.65 (<6)	8	1 (+2 at home)	55–60	Floor mat work	Group (2–3)	Certified Pilates physiotherapist	No	Yes
Fleming et al. [35], 2019	NA	<3 (PDDS)	8	2	60	Floor mat work	Individual	Certified Pilates instructor	No	Yes
Eftekhari and Etemadifar [36], 2018	RRMS (13)	2–6	8	3	50–60	Floor mat work	NA	NA	NA	Yes
Ozkul tel al. [37], 2018	RRMS	1 (0.87–2.12)	8	3	60	Floor mat work	NA	Physiotherapist	Yes (3)	Yes
Duff et al. [38], 2018	RRMS (14)/PPMS (1)	2.1 ± 1.8 (range 0–5, PDDS)	12	2	50	Apparatus work and floor mat work	Group (5–10)	Certified Pilates instructor	No	No
Eftekhari and Etemadifar [39], 2018	RRMS(13)	2–6	8	3	40–50	Floor mat work	NA	NA	NA	Yes
Kalron et al. [40], 2017	RRMS (22)	4.3 ± 1.3 (3–6)	12	1	30	NA	Individual	Certified Pilates physiotherapist	No	No
Bulguroglu et al. [41], 2017	NA	<4.5	8	2	60–90	Floor mat work or Reformer work	Individual	Certified Pilates physiotherapist	NA	No
Kara et al. [42], 2017	RRMS (9)	2.85 ± 1.57 (≤6)	8	2	45–60	Floor mat work	NA	Physiotherapist	Yes (4)	Yes
Fox et al. [43], 2016	RRMS (13/PPMS (12)/SPMS (8)	4–6.5	12	1	30	Floor mat work	Individual	Certified Pilates physiotherapist	No	Yes (in a previous study)
Küçük et al. [44], 2016	NA	3.2 ± 2.2 (≤6)	8	2	45–60	Floor mat work	Group	Physiotherapist	NA	Yes
Hosseini Sisi et al. [45], 2014	NA	0–4	8	3	60	NA	NA	NA	NA	No
Guclu-Cunduz et al. [46], 2014	NA	2 (0–4)	8	2	60	NA	Group	Certified Pilates physiotherapist	No	No
Marandi et al. [47], 2013	NA	<4.5	12	3	60	NA	NA	NA	NA	No

**EDSS**: Expanded Disability Status Scale; **min**: minute; **MS**: multiple sclerosis; **NA**: not available; **PDDS**: Patient-Determined Disease Steps; **PPMS**: primary progressive multiple sclerosis; **RRMS**: relapsing-remitting multiple sclerosis; **sd**: standard deviation; **SPMS**: secondary progressive multiple sclerosis.

**Table 3 jcm-11-00683-t003:** Methodological quality assessment of the reviewed studies using PEDro scale, and sample size calculation.

Study and Year	Sample Size Calculation	#1	#2	#3	#4	#5	#6	#7	#8	#9	#10	#11	Total
Güngör et al. [28], 2021	Yes	1	1	1	1	1	0	0	0	0	1	1	6/10
Fleming et al. [29], 2021	Yes	1	1	0	1	0	0	1	1	1	1	1	7/10
Gheitasi et al. [30], 2021	Yes	1	1	1	1	0	0	0	1	1	1	1	7/10
Arntzen et al. [31], 2020	Yes	1	1	1	1	0	0	1	1	1	1	1	8/10
Ozkul et al. [32], 2020	Yes	0	1	1	1	0	0	1	1	1	1	1	8/10
Banitalebi et al. [33], 2020	No	1	1	1	1	0	0	1	1	0	1	0	6/10
Abasiyanik et al. [34], 2020	Yes	1	1	0	1	0	0	0	0	0	1	1	4/10
Fleming et al. [35], 2019	No	1	1	1	1	0	0	0	0	0	0	1	4/10
Eftekhari and Etemadifar [36], 2018	No	1	1	0	1	0	0	1	0	0	1	1	5/10
Ozkul tel al. [37], 2018	Yes	1	1	1	1	0	0	1	1	0	0	1	6/10
Duff et al. [38], 2018	Yes	1	1	0	1	0	0	1	1	1	1	1	7/10
Eftekhari and Etemadifar [39], 2018	No	1	1	0	1	0	0	0	0	0	1	1	4/10
Kalron et al. [40], 2017	No	1	1	1	1	0	0	1	1	0	1	1	7/10
Bulguroglu et al. [41], 2017	No	1	1	0	1	0	0	1	0	0	1	1	5/10
Kara et al. [42], 2017	No	1	0	0	0	0	0	1	0	0	1	1	3/10
Fox et al. [43], 2016	Yes	1	1	1	1	0	0	1	1	1	1	1	8/10
Küçük et al. [44], 2016	No	1	1	0	1	0	0	0	1	0	1	1	5/10
Hosseini Sisi et al. [45], 2014	No	1	1	0	0	0	0	0	0	0	1	1	3/10
Guclu-Cunduz et al. [46], 2014	No	1	0	0	1	0	0	1	1	1	0	1	5/10
Marandi et al. [47], 2013	No	1	1	0	0	0	0	0	0	0	1	1	3/10
		17	16	9	15	1	0	10	10	6	15	17	

**#1**, eligibility criteria (not included in the total score); **#2**, random allocation; **#3**, concealed allocation; **#4**, baseline comparability; **#5**, participant blinding; **#6**, therapist blinding; **#7**, assessor blinding; **#8**, outcomes were obtained from more than 85%; **#9**, intention to treat analysis; **#10**, between-group difference; **#11**, point estimates and variability.

## Data Availability

All relevant data are included in the paper.

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
