# Peer review of "Therapeutic Effects of the Pilates Method in Patients with Multiple Sclerosis: A Systematic Review"

_jcm, 2022, doi:10.3390/jcm11030683_

Round 1

Reviewer 1 Report

  1. I would suggest to pay attention to some grammar issues (e.g. commas between subjects and verbs, repetitions or major grammar misunderstandings like "assessment of risk WAS been carried out"); it's important since the reading is not fluid and the argumentations result intricate;
  2. I would suggest to deepen the introduction and, in particular, to better specify the different clinical phenotypes of MS (progressive-relapsing MS does not exist anymore; see Lublin FD et al Defining the clinical course of multiple sclerosis: the 2013 revisions.) and the most prevalent symptoms of MS;
  3. Probably, even the hetherogeneity of the scales used along all the studies represent an important limit.

Author Response

Reviewer #1:

Q1.1. I would suggest to pay attention to some grammar issues (e.g. commas between subjects and verbs, repetitions or major grammar misunderstandings like "assessment of risk WAS been carried out"); it's important since the reading is not fluid and the argumentations result intricate;

A1.1. Thank you for bringing this to our attention. The manuscript has been completely revised and the technical writing of the manuscript has been improved, taking into account the reviewer's considerations.

Q1.2. I would suggest to deepen the introduction and, in particular, to better specify the different clinical phenotypes of MS (progressive-relapsing MS does not exist anymore; see Lublin FD et al Defining the clinical course of multiple sclerosis: the 2013 revisions.) and the most prevalent symptoms of MS;

A1.2 We appreciate the reviewer's comments. While Lublin et al. 2014 recommend modifying the classification of clinical MS phenotypes, most of the published studies still use the classic 1996 classification established by the US National Multiple Sclerosis Society (NMSS). This is indicated by de Cochrane review (Amatya et al. 2019) included in our review. To accommodate the reviewer´s suggestion, we have incorporated text in the introduction section (lines 43-46) summarizing the considerations indicated by Lublin et al. 2014, but we have retained the references to older classification systems. The most prevalent symptoms in MS are described in the text but they are not the main objective of our review, since they depend on disease stage and comorbidities; we focused on the symptoms indicated by each study, review and meta-analysis in our study.

Q1.3. Probably, even the hetherogeneity of the scales used along all the studies represent an important limit.

A1.3 We certainly agree with this suggestion, as we repeatedly highlighted heterogeneity of the scales in the discussion section. We have now incorporated text in the limitations section (lines 416-418) to further emphasize this limitation, following the reviewer’s suggestion.

Reviewer 2 Report

The study is of relevance and the methods are well described, nevertheless, I think that the results section could be improved.

Namely, table 1 is not easy to read, it has a lot of information and I feel that it loses its purpose (summarize and compare main findings). The reader has to go through each of the 20 studies instead of getting an overall picture. It also uses a lot of abbreviations. As you already have a detailed description on supplementary material, I suggest you to group the information of the  'findings' column more in accordance with the division made in the discussion section (ex. balance, fatigue,...) and use an organization system similar to the one used in table 2

Author Response

Q2.1. The study is of relevance and the methods are well described, nevertheless, I think that the results section could be improved.

A2.1. We appreciate the reviewer's comments.

Q2.2. Namely, table 1 is not easy to read, it has a lot of information and I feel that it loses its purpose (summarize and compare main findings). The reader has to go through each of the 20 studies instead of getting an overall picture. It also uses a lot of abbreviations. As you already have a detailed description on supplementary material, I suggest you to group the information of the ‘findings' column more in accordance with the division made in the discussion section (ex. balance, fatigue,...) and use an organization system similar to the one used in table 2.

A2.2 Thank you for bringing this to our attention. Table 1 has been completely revised and the information has been summarized. We are confident that the updated table is easier to read and has substantially improved.
